# Neutrophil–Galectin-9 Axis Linking Innate and Adaptive Immunity in ATL, Sézary Syndrome, COVID-19, and Psoriasis: An AI-Assisted Integrative Review

**DOI:** 10.3390/reports9010016

**Published:** 2025-12-31

**Authors:** Toshio Hattori

**Affiliations:** 1Roken Nursing Home, Akane, Kurashiki 712-8057, Japan; toshio.hattori.a3@tohoku.ac.jp; 2Graduate School of Public Health, Shizuoka Graduate University of Public Health, Shizuoka 420-0881, Japan

**Keywords:** ATL, Sézary syndrome, COVID-19, psoriasis, neutrophil, Galectin-9, T-cell receptor

## Abstract

Beyond their traditional role as short-lived antimicrobial cells, neutrophils are increasingly recognized as key regulators of adaptive immunity and tumor progression. This AI-assisted integrative review investigated the neutrophil–T-cell axis, particularly the role of Galectin-9 (Gal-9)*,* across adult T-cell leukemia/lymphoma (ATL), Sézary syndrome (SS), coronavirus disease 2019 (COVID-19), and psoriasis. Leveraging AI tools (GPT-5 and Adobe Acrobat AI Assistant) for literature synthesis (2000–2025) and expert validation, we aimed to identify common immunological mechanisms. Across all conditions, neutrophils displayed persistent activation, elevated Gal-9 expression, and modulated T-cell interactions. In ATL and SS, neutrophilia correlated with poor survival and TCR signaling dysregulation, suggesting Gal-9-mediated immune modulation. In COVID-19 and psoriasis, neutrophil-derived Gal-9-linked innate hyperactivation to T-cell exhaustion and IL-17-driven inflammation. These findings define a recurring neutrophil–Gal-9 regulatory module connecting innate and adaptive immune responses. This study underscores the feasibility of combining AI-driven literature synthesis with expert review to identify unifying immunological mechanisms and therapeutic targets across malignancy and inflammation.

## 1. Introduction

Neutrophils, historically viewed as short-lived innate effector cells, primarily responsible for pathogen defense are recognized for their broader functional spectrum. Recent immunological advances highlight their critical roles in modulating adaptive immunity, shaping the tumor microenvironment and contributing to disease progression [1]. Beyond their antimicrobial roles, neutrophils dynamically interact with T cells via cytokines, chemokines, and direct cell–cell contacts, thereby influencing immune responses in both malignant and inflammatory contexts. They also play key roles in supporting cancer metastasis through immune modulation and tissue remodeling [2,3]. Neutrophils can present antigens to T cells through the immune recognition system and promote the tumor-suppressive effect of CD8+ T cells [2]. They contribute to cancer initiation, growth, metastasis, and recurrence through mechanisms such as DNA damage, angiogenesis, immunosuppression, and neutrophil extracellular traps (NETs) [3].

Adult T-cell leukemia/lymphoma (ATL) and Sézary syndrome (SS) are prototypical T-cell malignancies characterized by dysregulated immune responses in which neutrophils are implicated [4,5]. In indolent ATL, patients with neutrophilia have significantly shorter survival than those with low neutrophil counts [6]. Similarly, in SS, neutrophil-mediated immune dysregulation is thought to exacerbate the immunosuppressive tumor microenvironment [7]. Inflammatory disorders such as coronavirus disease 2019 (COVID-19) and psoriasis are likewise characterized by prominent neutrophil infiltration and the formation of neutrophil extracellular traps (NETs), which drive tissue damage and chronic immune activation [8,9]. These diverse diseases, despite their distinct etiologies, share sustained neutrophil activation and altered T-cell immunity, making them pertinent models to investigate the neutrophil–galectin-9 (Gal-9) axis as a potential common denominator. This integrative review focuses explicitly on the four conditions, given emerging evidence suggesting a critical, yet complex, role for Gal-9 in mediating neutrophil–T cell cross-talk and disease pathology across both malignant and inflammatory contexts [10]. Gal-9 modulates T-cell receptor (TCR) signaling, promotes apoptosis of activated T cells, and alters neutrophil survival and migration, thereby linking innate and adaptive immunity. Gal-9 facilitates neutrophil adhesion to the vascular endothelium through CD44- and β2-integrin-dependent mechanisms, thereby enhancing their transmigration into inflamed tissues [11]. Collectively, these actions form a neutrophil–T-cell–galectin axis that appears to be a shared feature of both inflammatory and neoplastic disorders.

Despite numerous disease-specific studies, a comprehensive synthesis of the neutrophil–Gal-9 axis across distinct pathological settings remains limited. In this study, we used an artificial intelligence (AI)-assisted literature analysis to systematically explore neutrophil and galectin interactions in ATL, SS, COVID-19, and psoriasis. By integrating AI-based text mining with expert review, we sought to identify shared and disease-specific mechanisms that link innate activation, immune suppression, and tumor progression. This work not only highlights emerging immunoregulatory pathways but also demonstrates the value of AI-assisted analysis for integrating biomedical knowledge. A major strength of this study is its innovative methodology, in which AI-assisted literature synthesis accelerates data integration while preserving expert-driven interpretation and validation

## 2. Materials and Methods

### 2.1. Literature Search and AI Assistance

A comprehensive literature search was conducted to examine the roles of neutrophils, T-cell interactions, and galectin family proteins in ATL, SS, COVID-19, and psoriasis. The databases PubMed, Web of Science, and Scopus were queried for publications from January 2000 to August 2025. Keywords included “neutrophils”, “adult T-cell leukemia”, “Sézary syndrome”, “COVID-19”, “psoriasis”, “galectin-9”, and “*immune regulation*”. These diseases were selected because each has published evidence linking neutrophil activation to T-cell stimulation, yet their mechanistic intersections have not been reviewed together. It is also known that Gal-9 correlates with the pathology of the three diseases, excluding SS.

To assist with data synthesis, an AI tool (ChatGPT, GPT-5; OpenAI, San Francisco, CA, USA) was used to organize the preliminary literature. This initial search yielded approximately 312 relevant abstracts and titles (Table 1). The model was primarily used to process abstracts to generate summaries, cluster mechanistic themes, and highlight overlapping pathways across disease entities.

Summaries of individual references were generated using Adobe Acrobat AI Assistant for additional consistency and accuracy. This AI-assisted approach facilitated rapid identification of initial patterns and connections that might be overlooked or time-consuming in traditional manual reviews, offering an exploratory advantage over conventional meta-analysis by synthesizing diverse qualitative data. All AI-generated summaries were then rigorously verified, corrected, and refined by me using the full text of the original literature sources where available. In the process, it was found that there were no papers on Gal-9 in SS, but it has been revealed that SS, like ATL cells, is a tumor of CD4 cells; it shows clinical similarities to ATL, such as skin infiltration, and analyses such as TCR downregulation are very similar to those in ATL. I arbitrarily selected important papers outside the search and used them in the analysis [4,10,12].

Our AI-assisted method is not a replacement for meta-analysis when quantitative effect estimates are needed. Instead, it is particularly useful when the aim is to map mechanistic landscapes across heterogeneous models and endpoints, where pooling effect sizes is often not feasible.

### 2.2. Inclusion and Exclusion Criteria

We included peer-reviewed original research, clinical studies, case reports, and reviews published in English that examined neutrophil biology, T-cell regulation, or galectin-mediated mechanisms within the target diseases. Articles without direct relevance, non-English publications, or inaccessible full texts were excluded from analysis.

### 2.3. Data Synthesis

AI-assisted clustering results were integrated with my manual thematic analysis. Mechanistic overlaps, shared signaling patterns, and disease-specific differences were extracted and synthesized narratively. The author resolved discrepancies between automated and manual interpretations. Two figures and one table that reflect the main points of the paper were created by ChatGPT.

## 3. Results and Discussion

### 3.1. Disease-Specific Features of the Neutrophil–Galectin-9 Axis

ATL and Neutrophils

In blood smears from patients with ATL, neutrophils are frequently observed intermingled with leukemic cells, reflecting both tumor-driven inflammation and host immune activation [4]. One plausible mechanism involves cytokine overproduction induced by the Human T cell leukemia type 1–encoded *Tax* gene, which drives a pro-inflammatory milieu and may lead to neutrophilia. Opportunistic infections—common in ATL due to profound immunosuppression—further amplify neutrophil proliferation and recruitment. It also appears to be amplified by IL-8 produced by ATL cells [5]. In a long-term cohort of indolent ATL, patients with neutrophilia (≥7.5 × 10^9^/L) had significantly worse survival than those with lower counts (<7.5 × 10^9^/L) [6], highlighting the prognostic impact of sustained neutrophil activation. The precise causal mechanisms linking neutrophilia to poor survival and its specific interplay with Gal-9 require further mechanistic elucidation beyond observed associations.

Another layer of complexity arises from the downregulation of the T-cell receptor (TCR) in ATL cells, which may indirectly enhance neutrophil activity by altering cytokine signaling [12]. Genomic profiling has revealed persistent TCR pathway stimulation despite receptor downregulation, suggesting a chronic, noncanonical activation mechanism whose precise triggers remain unclear [12,13].

Gal-9, a β-galactoside-binding lectin, is markedly elevated in ATL sera before chemotherapy and declines in parallel with soluble interleukin-2 receptor (sIL-2R) levels after treatment, suggesting a relationship with tumor burden [14]. Expressed by both leukemic and stromal cells, Gal-9 engages the immune checkpoint receptor Tim-3, mediating immunoregulatory and potentially protumor effects [15]. These findings suggest that Gal-9 may contribute to ATL cell activation in vivo, serving as a functional bridge between innate immunity (neutrophil and myeloid activity) and leukemic T-cell signaling, thereby downregulating TCR expression in ATL cells. While these findings suggest a potential role for Gal-9 in ATL cell activation in vivo and in mediating crosstalk between innate immunity (neutrophil and myeloid activity) and leukemic T-cell signaling, potentially leading to TCR expression downregulation in ATL cells, further studies are needed to establish causality firmly.

The overarching conceptual framework for the neutrophil–Gal-9 axis across these disorders is illustrated in Figure 1, with ATL-specific details in Table 2.

### 3.2. Neutrophil Dysfunction and Immune Dysregulation in SS and ATL

In SS, peripheral neutrophil counts are often elevated, yet polymorphonuclear neutrophils (PMNs) exhibit impaired antimicrobial function, including reduced phagocytosis and intracellular killing of *Klebsiella pneumoniae* [16]. Phenotypically, SS PMNs display increased expression of CD11b and CD66b, accompanied by a concomitant loss of CD62L, reflecting a chronically activated yet functionally exhausted phenotype. Transcriptomic profiling reveals downregulation of TCR-associated genes such as *THEMIS* and *LAIR1*, paralleling patterns observed in ATL [16,17]. This T-cell dysfunction parallels patterns observed in ATL, suggesting convergent mechanisms of immune evasion and dysregulation across both malignancies, in addition to the fact that the tumor cells are CD4-positive cells. In SS, T cells in the tumor microenvironment exhibit a shift from a Th1 to a Th2 cytokine profile as the disease progresses. Malignant T cells show increased expression of Th2 cytokines (e.g., IL-4, IL-5, IL-10, IL-13). Additionally, tumor T cells express high levels of immune checkpoint molecules, contributing to immune suppression. Gal-9 may play a critical role in SS pathogenesis and progression [18]. While ATL’s neutrophilia primarily correlates with poor survival, SS presents a distinct scenario in which neutrophils are both numerous and functionally impaired, indicating nuanced differences in neutrophil activation and regulation between these two T-cell malignancies.

### 3.3. The Role of Gal-9 as a ‘Diagnostic and Prognostic Biomarker’ in COVID-19

Neutrophils play a dual and context-dependent role during SARS-CoV-2 infection. While they contribute to antiviral defense through degranulation, cytokine secretion, and the formation of neutrophil extracellular traps (NETs), excessive or dysregulated activation can promote tissue injury, vascular thrombosis, and acute respiratory distress syndrome. This paradoxical activity underscores the contribution of neutrophil-mediated hyperinflammation to COVID-19 pathogenesis [19]. Serum Gal-9 levels increase in COVID-19 and correlate positively with markers of systemic inflammation and sepsis (sTREM-1, MCP-1, IL-6, neutrophil-to-lymphocyte ratio, platelet-to-lymphocyte ratio, and erythrocyte sedimentation rate) [20]. These correlations suggest that elevated Gal-9 levels are potential diagnostic and prognostic biomarkers of hyperinflammatory states and disease severity in COVID-19, including mortality.

Beyond its utility as a biomarker, functional studies revealed that neutrophils isolated from COVID-19 patients secreted significantly higher levels of Gal-9 than those from healthy controls, irrespective of disease severity. Upon activation by SARS-CoV-2 or inflammatory stimuli, neutrophils shed surface-bound Gal-9, contributing to elevated plasma levels [8]. This neutrophil-derived Gal-9 is hypothesized to play an effector role in disease progression. Elevated extracellular Gal-9 correlates with T-cell exhaustion and reduced CD3–TCR expression, suggesting that neutrophil-derived Gal-9 may promote impaired adaptive immune responses (Figure 1). Together, these data define a “neutrophil–Gal-9 axis” that links innate overactivation with adaptive immune suppression in severe COVID-19.

### 3.4. Neutrophil–Galectin-9 Axis in Psoriasis

Neutrophils are abundant in psoriatic skin lesions and in peripheral blood, where their activation contributes to both the initiation and the perpetuation of disease [21]. A decline in circulating neutrophils parallels clinical improvement, indicating their active recruitment to inflammatory sites. NETs formation by psoriatic neutrophils promotes immune activation, autoantigen exposure, keratinocyte hyperproliferation, angiogenesis, and thrombosis. Through NETosis, neutrophils can also stimulate IFN-α production, enhance B-cell autoantibody generation, and drive IL-17-dependent T-cell activation—amplifying the inflammatory cascade characteristic of psoriasis [9].

Plasma Gal-9 levels are significantly elevated in psoriasis and correlate with inflammatory and immune checkpoint molecules [22]. In lesional skin, *LGALS9* expression is positively associated with cytokines such as IL-36B, IL-17RA, IL-6R, IL-10, and TGF-β1, suggesting Gal-9 participates in cytokine network regulation and immune homeostasis. Expanding on the role of T-cell stimulation by Gal-9, it is important to consider how Gal-9 influences the IL-17 axis, which is fundamental in psoriasis. IL-17 initiates and perpetuates the inflammatory response by acting on keratinocytes to cause cell proliferation and the production of pro-inflammatory molecules. This cytokine is primarily produced by distinct T-cell subsets including Th17 cells, γδ T cells, and innate lymphoid cells type 3 (ILC3) [9,21]. In this context, Gal-9, potentially derived from activated neutrophils, can engage immune checkpoint molecules such as Tim-3 on these T-cell populations, thereby influencing their activation state and cytokine production. This immune checkpoint modulation by Gal-9 likely plays a role in sustaining or exacerbating skin inflammation by altering the balance of pro-inflammatory (e.g., IL-17) and regulatory responses, thereby reinforcing the pathological neutrophil–T-cell axis in psoriasis. The precise interplay between neutrophil-derived Gal-9 and these specific IL-17-producing cells warrants further investigation to elucidate its contribution to psoriasis pathogenesis fully.

Although correlations with total white blood cell count, eosinophil percentage, and alanin aminotransferase are modest, elevated Gal-9 likely reflects systemic inflammatory activity and disease burden.

Table 2 provides a comparative overview of neutrophil activation status, galectin-9 sources and levels, and their effects on T-cell receptor signaling across the four conditions. (Table 2)

Figure 1 schematically summarizes these findings, illustrating how neutrophil-derived or tissue-derived galectin-9 differentially modulates T-cell activation and dysfunction in a disease-dependent manner.

This schematic summarizes disease-specific patterns of neutrophil activation, galectin-9 (Gal-9) expression, and their effects on T-cell receptor (TCR) signaling across adult T-cell leukemia/lymphoma (ATL), Sézary syndrome (SS), COVID-19, and psoriasis.

In ATL, absolute neutrophilia is frequently observed and is associated with poor prognosis. Leukemic CD4^+^ T cells exhibit downregulation of surface CD3/TCR expression despite persistent intracellular TCR pathway activation. Gal-9 is expressed by ATL cells and stromal components and is elevated in patient plasma, suggesting a role in immune modulation and leukemic T-cell signaling.

In Sézary syndrome, circulating malignant CD4^+^ T cells show reduced expression of TCR-associated signaling molecules, accompanied by functional immune dysregulation. Neutrophils are often increased in number but display impaired antimicrobial function, indicating a chronically activated yet dysfunctional phenotype. Direct evidence for Gal-9 involvement remains limited, but shared T-cell signaling abnormalities suggest a mechanistic overlap with ATL.

In COVID-19, excessive neutrophil activation and NET formation contribute to systemic inflammation and tissue damage. Activated neutrophils represent a major source of circulating Gal-9, which correlates with disease severity and inflammatory markers. Elevated Gal-9 is associated with T-cell exhaustion, linking innate hyperactivation to adaptive immune suppression.

In **psoriasis**, neutrophils accumulate in skin lesions and peripheral blood and promote IL-17-driven inflammation and cytokine amplification. Gal-9 expression is increased in lesional tissues and circulation and is associated with inflammatory and immune checkpoint pathways, suggesting context-dependent modulation of T-cell activation rather than direct TCR downregulation.

Overall, the figure illustrates Gal-9 as a recurring but context-dependent immunomodulator within the neutrophil–T-cell axis, highlighting both shared themes and disease-specific heterogeneity.

### 3.5. Context-Dependent T-Cell Stimulation by Galectin-9

Gal-9 exerts multifaceted and context-dependent effects on T-cell signaling and fate [23,24]. Upon T-cell activation, intracellular Gal-9 translocates to the plasma membrane and co-localizes with zeta chain-associated protein kinase 70 (ZAP70), promoting the phosphorylation of proximal TCR signaling molecules as shown in Figure 2. Mechanistically, these phosphorylation events activate endocytic machinery that removes TCR–CD3 complexes from the surface. This forms part of a self-limiting feedback loop that prevents overactivation and maintains immune homeostasis [25] (Figure 2).

This potentiates downstream activation and cytokine production. In Gal-9-deficient (Lgals9^−/−^) T cells, proximal TCR signaling and IL-17 production are markedly reduced; however, this defect can be rescued by phorbol 12-myristate 13-acetate (PMA)/ionomycin, indicating that Gal-9 primarily modulates early TCR signaling steps. Specifically, PMA stimulation activates protein kinase C (PKC), which promotes the surface expression of Gal-9, and PKC inhibitors partially suppress the PMA-induced increase in Gal-9 expression [26]. PKCα and PKCθ are the major PKC isotypes involved in TCR down-regulation, a critical process that regulates TCR surface expression and the immune response. PKC isotypes regulate immune responses and optimize T cell activation by down-regulating TCR signaling [27] (Figure 2).

## 4. Conclusions

Across ATL, Sézary syndrome, COVID-19, and psoriasis, our analysis suggests that Gal-9 functions as a recurring immunomodulator linking neutrophil activation to altered T-cell receptor (TCR) signaling. Rather than a conserved master regulator, Gal-9 appears to participate in disease-specific circuits that converge on neutrophil–T-cell interactions.

A notable paradox emerges: Gal-9 is detected in T-cell tumor tissue in specific contexts (e.g., ATL), yet it also contributes to T-cell exhaustion or functional impairment in others (e.g., chronic viral infection or inflammatory skin disease). This duality highlights the context-dependent nature of Gal-9 biology.

Despite consistent associative findings across multiple conditions, direct mechanistic evidence remains limited. Variability in neutrophil phenotypes, disease-specific cytokine environments, and differences in Gal-9 sources (tumor cells, neutrophils, keratinocytes) introduce heterogeneity that warrants careful interpretation. Further experimental validation is needed to disentangle whether Gal-9 primarily drives, amplifies, or merely reflects pathological immune states.

These observations lead to several testable hypotheses with potential translational impact. In ATL, targeted Gal-9 blockade or modulation of downstream TCR-signaling pathways may represent therapeutic avenues worth exploring. In psoriasis or COVID-19, interventions targeting Gal-9-dependent neutrophil activation or IL-17-driven inflammation could refine future treatment strategies. Ultimately, viewing Gal-9 as a dynamic component within a broader neutrophil–T-cell axis may enable more precise disease-specific therapeutic approaches.

## Figures and Tables

**Figure 1 reports-09-00016-f001:**
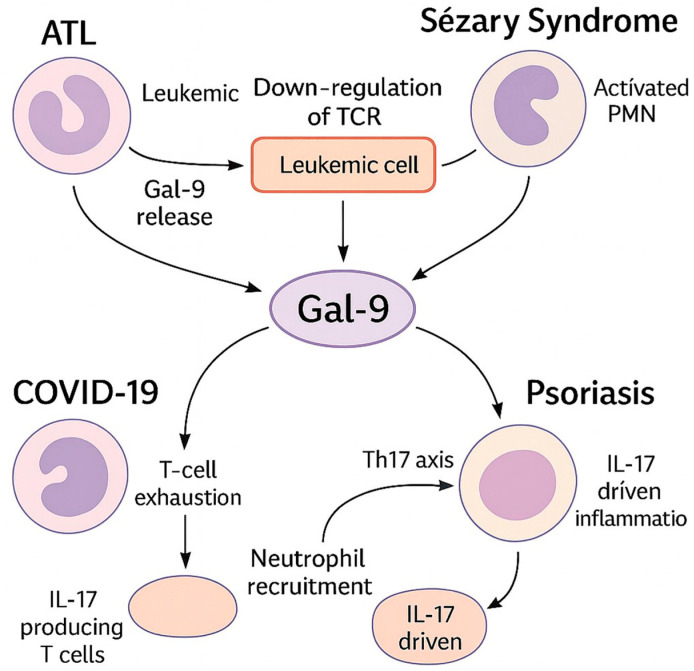
Involvement of neutrophils and the T-cell axis via galectin-9 in various diseases.

**Figure 2 reports-09-00016-f002:**
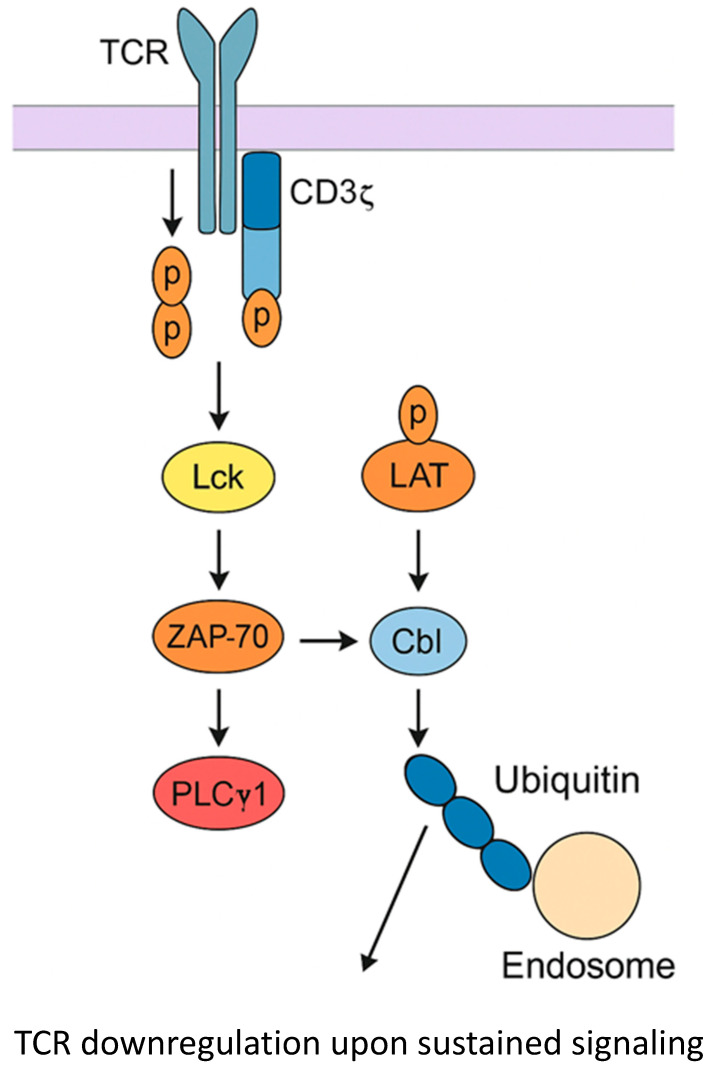
Stimulation of TCR signaling by Gal-9. Gal-9 colocalizes with ZAP70, which promotes the phosphorylation of light chain kinase (Lck), ZAP70, phospholipase Cγ1 (PLC γ1) and linker for activation of T cell (LAT). These events trigger recruitments of ubiquitin ligases (Cb1) and activate endocytic machinery that removes TCR-CD3 complexes from the surface [25].

**Table 1 reports-09-00016-t001:** AI-assisted literature selection workflow.

Step	Description	Number of Records
1. Initial AI-assisted retrieval	Comprehensive literature search using GPT-5 and Adobe Acrobat AI Assistant covering ATL, Sézary syndrome, COVID-19, psoriasis, neutrophils, Galectin-9, and TCR signaling.	312
2. Title and abstract screening	AI-based clustering followed by manual exclusion of irrelevant or non-mechanistic studies.	142
3. Full-text eligibility assessment	Evaluation of study design, disease relevance, and mechanistic contribution focusing on Gal-9, neutrophil biology, and TCR regulation.	67
4. Final inclusion for synthesis	Studies selected for integration into the mechanistic and conceptual framework of this integrative review.	20

**Table 2 reports-09-00016-t002:** Comparative overview of the neutrophil–Galectin-9 axis across representative inflammatory and malignant diseases.

Disease Context	Neutrophil Activation/Status	Primary Source of Galectin-9	Gal-9 Levels	Effects on T Cells/ATL	TCR Expression/Signaling	Innate–Adaptive Crosstalk	Clinical Correlations/Biomarkers	Key References
Adult T-cell leukemia/lymphoma (ATL)	Neutrophilia driven by Tax-mediated cytokine storm and infection-associated inflammation.	Leukemic + stromal cells; possible contribution from activated neutrophils.	High prior to therapy; declines with sIL-2R after chemotherapy.	Tim-3-mediated immunoregulation; potential driver of ATL activation.	Surface TCR downregulated; pathway chronically stimulated (noncanonical, cytokine- and Gal-9–associated).	Neutrophil-Gal-9 bridges myeloid activation and leukemic T-cell signaling.	Neutrophilia ≥ 7.5 × 10^9^/L predicts poor survival; Gal-9 correlates with tumor load.	[4,6,12,13,14,15]
Sézary syndrome (SS)	Elevated counts with reduced antimicrobial function; ↑CD11b, ↑CD66b, ↓CD62L phenotype.	Not clearly defined (limited direct evidence).	Not well quantified in SS-specific studies.	T-cell dysfunction with enhanced MDSC suppression.	Downregulation of THEMIS and LAIR1.	Aberrant neutrophil-T-cell signaling and MDSC-driven suppression.	Impaired microbicidal activity despite neutrophilia.	[16,17,18]
COVID-19	Hyperactivated neutrophils with NETosis; contribute to ARDS and thrombosis.	Activated neutrophils (major source).	Elevated; correlates with sTREM-1, MCP-1, IL-6, NLR, PLR, ESR.	Induces T-cell exhaustion and reduced CD3-TCR expression.	Downregulated CD3-TCR complex.	Neutrophil-derived Gal-9 links innate hyperactivation to adaptive suppression.	Severity and mortality correlate with Gal-9.	[8,19,20]
Psoriasis	Abundant NET-forming neutrophils; drive IL-17-dependent inflammation.	Lesional keratinocytes (LGALS9) and circulating cells.	Increased; correlates with inflammatory/checkpoint molecules.	Promotes IL-17-mediated T-cell activation.	TCR modulation is indirect via cytokine milieu.	NETs amplify Th17 and autoantigen responses.	Neutrophil decline parallels clinical remission.	[9,21,22]

Abbreviations: ATL, adult T-cell leukemia/lymphoma; SS, Sézary syndrome; Gal-9, galectin-9; MDSC, myeloid-derived suppressor cell; NETs, neutrophil extracellular traps; sIL-2R, soluble interleukin-2 receptor; ↑, enhanced expression; ↓, diminished expression.

## Data Availability

No new experimental data were generated in this study. All information presented was derived from previously published sources cited within the article. Figures and tables were generated using AI assistance (OpenAI GPT-5) based on these published data. The datasets underlying the referenced literature are available from the respective publishers or repositories.

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
