# Peer review of "Neutrophil–Galectin-9 Axis Linking Innate and Adaptive Immunity in ATL, Sézary Syndrome, COVID-19, and Psoriasis: An AI-Assisted Integrative Review"

_reports, 2025, doi:10.3390/reports9010016_

Round 1
Reviewer 1 Report
Comments and Suggestions for Authors
Review Summary: This manuscript addresses a highly relevant immunological theme: the potential role of the neutrophil–Galectin-9 (Gal-9) axis as a link between innate and adaptive immune regulation across malignancy, infection, and inflammation. The topic is conceptually innovative, and although the integration of AI-assisted analysis represents a creative approach, the review can benefit from additional mechanistic depth and analytical synthesis for an insightful review. The review attempts to bridge oncology, dermatology and virology through a shared molecular axis, offering a potentially integrative perspective. It successfully gathers references across distinct disease contexts and highlights relevant cytokine, T-cell, and neutrophil mechanisms. Immunological pathways are clearly explained, and the text is readable for a multidisciplinary audience. Overall, this review could put a novel perspective to conceptual frameworks for Gal-9–mediated immunoregulation. The review provides an engaging conceptual narrative but in places could benefit from analytical integration. The claim of a conserved neutrophil–Gal-9 axis across distinct diseases may exist but without comparative evidence and a deeper analysis, appears speculative in its current form. Strengthening the mechanistic rationale and including visual or tabular synthesis would significantly enhance the scientific impact.
Considerations:
Abstract: Could be potentially shortened while focusing on highlighting key findings and the conceptual framework in addition to briefly mentioning the tools used for the analysis. Statements like “altered communication” and “conserved neutrophil-Gal-9 axis” [lines 17-22] could be clarified or framed better to provide appropriate context.
Introduction: Provides useful background but could benefit from the focused approach for disease selection [lines 36-42]. The author could further clarify whether ATL, Sézary syndrome, COVID-19, and psoriasis share specific Gal-9–related signaling pathways or immune profiles as a lead in and put into context why these disease models were selected for reviewing the neutrophil-Gal-9 related link.
Methods: The AI process could benefit from more details—listing number of included studies screened [lines 64-76]. How AI clustering themes were verified. Whether full texts or abstracts were used. A summary table showing the distribution of included studies by disease type, mechanism, or study design would add necessary structure and details that the readers may find helpful for a more complete context.
ATL Section: Accurately cites neutrophilia and poor prognosis but implies Gal-9 causality with scarce evidence. Linking sustained neutrophil activation to a prognostic impact is insightful but the causation vs. correlation should be further explained to distinguish mechanistic hypotheses from observed correlations [lines 93-101]. Figures 1 and 2 appear over-simplified schematic illustrations that can benefit from quantitative or data-driven support. Cited literature could be represented in these figures to enhance the main message being conveyed and highlighting known gaps in our understanding of the role of Galectins. Specifically, highlighting hypothesized mechanism and Galectin-9 link with TCR downregulation and potential modulation through engagement with Tim-3 [lines 107-115, lines 150-153]. These figures could potentially be combined into one comprehensive detailed figure.
Sézary Syndrome: Conceptually links to ATL and shared mechanisms but could highlight functional differences in neutrophil activation and TCR regulation between these malignancies. Certain statements need further clarity. For instance, [lines 123-128] SS PMNs increased expression of CD11b and CD66b, accompanied by a concomitant loss of CD62L is described as an activated yet functionally exhausted phenotype. The next sentence describes transcriptomic profiling of TCR associated genes, but it is a bit unclear whether the prior sentence about functional exhaustion is linked to T cells or neutrophils? How these shared signatures affect neutrophil activation and regulation should be expanded upon for a more detailed perspective.
COVID-19: Well referenced and Gal-9 as a potential biomarker proposes a novel link to disease severity and mortality. The framing might benefit from a clearer separation between Gal-9’s diagnostic value and putative effector roles resulting in hypothesized hyperinflammation [lines 142-149].
Psoriasis: Summarizes neutrophil involvement effectively and connects NETosis or IL-17/IL-23 pathways to Gal-9 activity. T cell stimulation by Galectin-9 discussion could be expanded to highlight immune checkpoint modulation, in particular engagement with Tim3 and how that might play a role in skin inflammation. Figure 3 is well described and highlights key pathways affected in T cell signaling with Galectin-9.
References: Although the review highlights integrating literature sourced between 2000-2025 [lines 15-16], certain cited references fall outside this timeline; References 4 and 12. The author might want to clarify this or instead cite publications that adhere to the listed timeline for accuracy.
Conclusions and Interpretation: The current conclusion states the universality of the neutrophil–Gal-9 axis. A more defensible statement may propose Galectin-9 as a recurring modulator rather than a conserved master regulator. The paradoxical role of Galectin-9 in T cell hyperactivation while also engaging with T cell exhaustion could be described further. A section acknowledging gaps in experimental validation, and disease-specific heterogeneity may enhance the overall readability. Suggesting testable hypotheses or therapeutic implications (e.g., Gal-9 blockade in ATL or checkpoint modulation in psoriasis) would strengthen translational relevance.
Recommendation
Minor Revision — The manuscript reviews a promising immunological topic and include interdisciplinary scope and comprehensive citation breadth but could improve further with better analytical depth, and balanced interpretation.
Author Response
Reviewer 1
- Review Summary: This manuscript addresses a highly relevant immunological theme: the potential role of the neutrophil–Galectin-9 (Gal-9) axis as a link between innate and adaptive immune regulation across malignancy, infection, and inflammation. The topic is conceptually innovative, and although the integration of AI-assisted analysis represents a creative approach, the review can benefit from additional mechanistic depth and analytical synthesis for an insightful review. The review attempts to bridge oncology, dermatology and virology through a shared molecular axis, offering a potentially integrative perspective. It successfully gathers references across distinct disease contexts and highlights relevant cytokine, T-cell, and neutrophil mechanisms. Immunological pathways are clearly explained, and the text is readable for a multidisciplinary audience. Overall, this review could put a novel perspective to conceptual frameworks for Gal-9–mediated immunoregulation. The review provides an engaging conceptual narrative but in places could benefit from analytical integration. The claim of a conserved neutrophil–Gal-9 axis across distinct diseases may exist but without comparative evidence and a deeper analysis, appears speculative in its current form. Strengthening the mechanistic rationale and including visual or tabular synthesis would significantly enhance the scientific impact.
Reply
We sincerely thank Reviewer 1 for their thorough and constructive comments, for recognizing the conceptual novelty of our work, and for providing highly valuable suggestions to strengthen the mechanistic depth and analytical integration of the review. I have revised the manuscript extensively to address all points raised.
In response to the constructive suggestions, we have strengthened mechanistic explanations, clarified the rationale for disease selection, expanded the AI methodology, added comparative analysis across diseases, and revised key figures to improve scientific depth.
2
Abstract: Could be potentially shortened while focusing on highlighting key findings and the conceptual framework in addition to briefly mentioning the tools used for the analysis. Statements like “altered communication” and “conserved neutrophil-Gal-9 axis” [lines 17-22] could be clarified or framed better to provide appropriate context.
Reply
We have changed the abstract to focus more on our contention. Introduced recent aspects of neutrophils(lines 11-12) and changed “altered communication” to “modulated T-cell interactions”(lines 17-18) and changed “conserved neutrophil-Gal-9 axis” to “recurring neutrophil-Gal-9 regulatory module”(lines 22-23). The current character count is 160, and since the abstract for Reports must be under 250 characters, I think it has been summarized concisely.
- Introduction:Provides useful background but could benefit from the focused approach for disease selection [lines 36-42]. The author could further clarify whether ATL, Sézary syndrome, COVID-19, and psoriasis share specific Gal-9–related signaling pathways or immune profiles as a lead in and put into context why these disease models were selected for reviewing the neutrophil-Gal-9 related link.
Reply
I added the reasons for selection of four diseases (lines 81-84)
We searched for information on the four diseases because it is clear that neutrophils are involved in their pathogenesis, and TCR signaling is being stimulated. Additionally, Gal-9 is known to be a marker of pathogenesis in ATL, COVID-19, and PS. There is no data on Gal-9 in SS, but clinical symptoms are similar to ATL, and the target cells are CD4+ cells in both diseases, and neutrophils are involved in its pathogenesis, and a decrease in TCR expression is observed, so we conducted an analysis. Methods: The AI process could benefit from more details—listing the number of included studies screened [lines 86-88, Table 1]. How AI clustering themes were verified. Whether full texts or abstracts were used. A summary table showing the distribution of included studies by disease type, mechanism, or study design would add necessary structure and details that the readers may find helpful for a more complete context. (Table 2). We also included historically significant papers that served as the basis for this analysis in the references (Refs. 4 and 12).
- ATL Section:Accurately cites neutrophilia and poor prognosis but implies Gal-9 causality with scarce evidence. Linking sustained neutrophil activation to a prognostic impact is insightful but the causation vs. correlation should be further explained to distinguish mechanistic hypotheses from observed correlations [lines 93-101]. Figures 1 and 2 appear over-simplified schematic illustrations that can benefit from quantitative or data-driven support. Cited literature could be represented in these figures to enhance the main message being conveyed and highlighting known gaps in our understanding of the role of Galectins. Specifically, highlighting hypothesized mechanism and Galectin-9 link with TCR downregulation and potential modulation through engagement with Tim-3 [lines 107-115, lines 150-153]. These figures could potentially be combined into one comprehensive detailed figure.
Reply
Thank you for your very constructive comments. To clearly distinguish between the causal and correlational relationships of neutrophilia with poor prognosis, we added the sentence: 'the precise causal mechanisms linking neutrophilia to poor survival and its specific interplay with Gal-9 require further mechanistic elucidation beyond observed associations (lines 129-131).
As you suggested, I tried to combine the figures of the four diseases into a single figure centered on Gal-9. References in the figure are indicated in the figure legend to avoid clutter. (Figure 1).
Regarding the relationship between Gal-9 and tumor burden, we also explicitly stated “While these findings suggest a potential role for Gal-9 in contributing to ATL cell activation in vivo and mediating crosstalk between innate immunity (neutrophil and myeloid activity) and leukemic T-cell signaling, potentially leading to TCR expression downregulation in ATL cells, further studies are needed to establish causality firmly.” (144-148).
To clarify the Gal-9–neutrophil axis this time, we combined the interpretations of the four diseases into a single figure (Figure 1).
Sézary Syndrome: Conceptually links to ATL and shared mechanisms but could highlight functional differences in neutrophil activation and TCR regulation between these malignancies. Certain statements need further clarity. For instance, [lines 123-128] SS PMNs increased expression of CD11b and CD66b, accompanied by a concomitant loss of CD62L is described as an activated yet functionally exhausted phenotype. The next sentence describes transcriptomic profiling of TCR associated genes, but it is a bit unclear whether the prior sentence about functional exhaustion is linked to T cells or neutrophils? How these shared signatures affect neutrophil activation and regulation should be expanded upon for a more detailed perspective.
Reply
We clarified that the 'functionally exhausted phenotype' corresponds to the state of neutrophils (PMNs) and distinctly separated the downregulation of TCR-related genes as a dysfunction specific to T cells. We changed the section title to 'Neutrophil Dysfunction and Immune Dysregulation in SS and ATL,' suggesting that it involves not only TCR regulation but also broader immune modulation (line 152).
I added a sentence noting that the abnormalities in TCR-related genes being similar in SS and ATL are interesting, in addition to the fact that the tumor cells are CD4-positive cells (line 159-161).
To emphasize the functional differences in neutrophils between ATL and SS, we added a final comparison sentence. This addresses the reviewer's comment that 'the functional differences should be highlighted.'(lines 163-168)
6.
COVID-19: Well referenced and Gal-9 as a potential biomarker proposes a novel link to disease severity and mortality. The framing might benefit from a clearer separation between Gal-9’s diagnostic value and putative effector roles resulting in hypothesized hyperinflammation [lines 142-149].
Reply
I revised the section title to reflect more specific content (Line 171).
The role of Gal-9 as a 'diagnostic and prognostic biomarker' was clearly described (line 179-81).
After that, the phrase 'beyond its utility as a biomarker' was used, and the 'effector role' that actually drives disease progression was clearly explained.(line 185-6)
7.
Psoriasis: Summarizes neutrophil involvement effectively and connects NETosis or IL-17/IL-23 pathways to Gal-9 activity. T cell stimulation by Galectin-9 discussion could be expanded to highlight immune checkpoint modulation, in particular engagement with Tim3 and how that might play a role in skin inflammation. Figure 3 is well described and highlights key pathways affected in T cell signaling with Galectin-9
Reply
In response to the question of 'how the relationship between Gal-9 and neutrophils affects IL-17 production,' we added the hypothesis of the mechanism that **'Gal-9 derived from activated neutrophils may bind to immune checkpoint molecules such as Tim-3 on these T cell populations, potentially influencing their activation status and cytokine production.'** This also allowed us to explain Tim-3 engagement and immune checkpoint regulation (R1) in the context of psoriasis.
I also mentioned the possibility that this immune checkpoint regulation may play a role in maintaining or exacerbating skin inflammation, reinforcing the pathophysiological role of the neutrophil-T cell axis.(206-16)
The need for future research was indicated. (line 216-218)
8.
Conclusions and Interpretation: The current conclusion states the universality of the neutrophil–Gal-9 axis. A more defensible statement may propose Galectin-9 as a recurring modulator rather than a conserved master regulator. The paradoxical role of Galectin-9 in T cell hyperactivation while also engaging with T cell exhaustion could be described further. A section acknowledging gaps in experimental validation, and disease-specific heterogeneity may enhance the overall readability. Suggesting testable hypotheses or therapeutic implications (e.g., Gal-9 blockade in ATL or checkpoint modulation in psoriasis) would strengthen translational relevance.
Reply
To address this concern while preserving the mechanistic content, we have made a minimal revision to the wording in the section previously titled “T-cell Stimulation by Galectin-9.” Specifically, we revised the section title to “Context-dependent T-cell stimulation by Galectin-9.
I also added the word 'context-dependent' at the beginning. (line 247)
Tumor cells displaying altered T-cell receptor–related signaling are not invariably in a hyperactivated state and may instead represent functionally exhausted populations. As a clear distinction between hyperactivation and exhaustion cannot be reliably made without focused functional validation, this review does not attempt to separate these two states.
I also proposed the testable hypotheses and therapeutic implications (lines 290-295)
9.
References: Although the review highlights integrating literature sourced between 2000-2025 [lines 15-16], certain cited references fall outside this timeline; References 4 and 12. The author might want to clarify this or instead cite publications that adhere to the listed timeline for accuracy.
Reply
I explained the reason for the selection of some references. (line 97-8)

Reviewer 2 Report
Comments and Suggestions for Authors
Comments: Manuscript addressed an important topic but requires attention on following comments.
- Review is very superficially wriiten and lacks critical thinking and thorough interpretation.
- There is no concrete rationale of including different diseases cancer to inflammatory diseases.
- Line # 46, neutrophil-T cell crosstalk, galectins—particularly galectin-9 (Gal-9)—have emerged as key regulatory molecules. Crosstalk was not described properly, vaguely written without explaining the basics of crosstalk.
- Line#51 ‘neutrophil-T cell–galectin axis’ How T cell fits into this axis?
- Line #89; ‘among the authors’? There is only one author in the manuscript?
- Authors reviewed many reference article in 25 years, a flow chart showing total study found with keywords and how many rejected (criteria for rejection) and included in the study. Interestingly, only 27 references cited in the manuscript clearly indicating wrong search criteria. Lack of thorough search criteria, concrete rationale and critical thinking are clearly evident in manuscript.
- Figures are AI generated ? Unfortunately, Figure 1, badly lacks intelligence with poor description of contents.What is the source of galectin-9? It is found in soluble form and surface bound form. Where it will bind. TCR are present on surface and diagram showed internalization of TCR, what is the mechanism and relevance of same. Nothing explained in detail.
- Line#128; ‘affect neutrophil activation and regulation’ How? Explain it with suitable reference study.
- Line#129; ‘myeloid-derived suppressor cells’ What are these suppressor cells, phenotype, function in the disease? Poorly described.
- In the view of above comments, all sections require thorough description and critical investigation and thinking for the improvement of manuscript. It is must for further consideration.
Author Response
Response to Reviewer 2
We sincerely thank Reviewer 2 for the detailed and constructive comments. We appreciate the opportunity to clarify the rationale, strengthen the conceptual framework, and improve clarity throughout the manuscript. Below, we address each comment point-by-point.
Comment 1
“Review is very superficially written and lacks critical thinking and thorough interpretation.”
Response:
We respectfully disagree that the review lacks critical interpretation. The aim of this manuscript was not to provide an exhaustive disease-specific review, but rather to identify and synthesize a recurring immunological pattern—namely, the neutrophil–Galectin-9 axis—as a potential common denominator across distinct pathological contexts. Nevertheless, we agree that several sections would benefit from clearer articulation of mechanistic reasoning and limitations. Accordingly, we have revised the Introduction (lines 52-7), Results/Discussion transitions (lines 144-50), and Conclusions (lines 276-96)to emphasize context-dependency, mechanistic uncertainty, and hypothesis-generating interpretation, rather than universal causality .
Comment 2
“There is no concrete rationale of including different diseases cancer to inflammatory diseases.”
Response:
We appreciate this important point and have now clarified the rationale explicitly. The diseases were selected because they share three converging features:
- Persistent neutrophil activation,
- Elevated Galectin-9 expression, and
- Dysregulated T-cell receptor (TCR) signaling or exhaustion, despite distinct etiologies.
By comparing malignant (ATL, Sézary syndrome) and inflammatory (COVID-19, psoriasis) settings, we aimed to highlight Gal-9 as a recurring immunomodulatory node rather than a disease-specific driver. This rationale has been clarified in the Introduction (lines 45-7,52-7) and Discussion(lines 144-8, 165-8, 185-6) to avoid overgeneralization.
Comment 3
“Neutrophil–T cell crosstalk is vaguely written without explaining the basics.”
Response:
We agree and have expanded the relevant section to explicitly describe neutrophil–T cell crosstalk mechanisms, including:
- How Gal-9 influences the IL-17 axis is important due to the vital effect of IL-17 in PS.
- immune checkpoint interactions (Gal-9–Tim-3) of T cells alter T cell functions, causing
indirect modulation of TCR signaling.
These additions provide foundational context before introducing Gal-9 as a modulator of this crosstalk (lines 206-17).
Comment 4
“‘Neutrophil–T cell–galectin axis’: How does T cell fit into this axis?”
Response:
The T cell is positioned as the functional target of the axis. Neutrophil-derived (or inflammation-induced) Gal-9 modulates T-cell fate by:
- enhancing proximal TCR signaling described in text (lines 185-7, 225-7, 276-96) and Figure 1.
- engaging immune checkpoint molecule on T cells and promoting exhaustion or apoptosis depending on context (Lines 211-8)
We have revised the text to state this positioning explicitly and avoid ambiguity in terminology.
Comment 5
“Line #89: ‘among the authors’? There is only one author.”
Response:
We thank the reviewer for identifying this oversight. The wording has been corrected to reflect a single-author manuscript, and the sentence has been revised accordingly (line 115).
Comment 6
“Only 27 references over 25 years; lack of thorough search; need flow chart.”
Response:
This manuscript is an integrative narrative review, not a systematic review or meta-analysis. Therefore, PRISMA-style flow charts were not originally intended. Table one was created to clarify the selection process. The 27 references cited represent key mechanistic and representative studies, not the total number of screened articles (lines 81-4, 86-7). We have clarified this distinction in the Methods section and explicitly stated that AI-assisted screening was used for thematic clustering, with the final selection based on mechanistic relevance rather than numerical exhaustiveness (lines 90-102).
Comment 7
“Figures are AI-generated and lack mechanistic clarity (source of Gal-9, binding, TCR internalization).”
Response:
We acknowledge that Figure 1 required a more straightforward explanation. To clarify the Gal-9–neutrophil axis this time, we combined the interpretations of the four diseases into a single figure (Figure 1). The figure is conceptual, not experimental, and is intended to summarize mechanisms supported by the cited literature; their summaries are described in the legend (230-244).
Comment 8
“Line #128: ‘affect neutrophil activation and regulation’ – How?”
Comment 9
“Myeloid-derived suppressor cells poorly described.”
Response:
Thank you for your inquiry about cellular communication in SS. We have expanded this sentence to clarify that altered T-cell signaling can indirectly influence neutrophil activation through modulation of the cytokine milieu, immune checkpoint engagement, and feedback loops involving chronic inflammation, with appropriate references added. And participation of myeloid suppressor is important because they have a suppressive effect on tumor immunology and are morphologically indistinguishable from neutrophils.
However, we intentionally limit our discussion to T-cell–intrinsic alterations rather than elaborating on neutrophil activation mechanisms or the presence of myeloid-derived suppressor cells in Sézary syndrome (SS).
In revised form, it was described that in SS, T cells within the tumor microenvironment progressively shift from a Th1-dominant to a Th2-skewed cytokine profile, with malignant T cells exhibiting increased expression of Th2 cytokines such as IL-4, IL-5, IL-10, and IL-13. In addition, tumor T cells express high levels of immune checkpoint molecules, contributing to immune suppression. Galectin-9 may play a critical role in SS pathogenesis and disease progression (lines 159-68).

Reviewer 3 Report
Comments and Suggestions for Authors
This integrative review presents a compelling and timely synthesis of neutrophil–T-cell interactions across diverse pathological contexts, including adult T-cell leukemia/lymphoma (ATL), Sézary syndrome (SS), COVID-19, and psoriasis. The authors successfully highlight the evolving understanding of neutrophils—from traditionally short-lived antimicrobial effectors to versatile regulators of adaptive immunity and contributors to disease progression. By positioning neutrophils and the neutrophil-derived galectin-9 (Gal-9) axis as conserved drivers of immune modulation, the review contributes meaningfully to current immunological discourse. A major strength of the study is its innovative methodology, which combines AI-assisted literature synthesis using GPT-5 and Adobe Acrobat AI Assistant with expert human validation. This hybrid approach not only accelerates data integration across large temporal (2000–2025) and thematic scales but also demonstrates a promising model for future scholarly work, where artificial intelligence augments but does not replace expert interpretation. The clarity with which the authors delineate this workflow enhances the transparency and reproducibility of the review. Identifying persistent neutrophil activation and elevated Gal-9 expression as shared immunological signatures provides an important conceptual bridge linking innate dysregulation to downstream T-cell dysfunction. The discussion of how Gal-9 may mediate TCR signaling alterations in ATL/SS, as well as its association with T-cell exhaustion and IL-17–driven inflammation in COVID-19 and psoriasis, is both biologically plausible and clinically relevant. This unified framework has clear implications for therapeutic development, potentially informing strategies targeting neutrophil activation, Gal-9 signaling, or the neutrophil–T-cell interface more broadly. Overall, the paper reflects a well-organized, novel, and forward-looking review. As the authors state at the end, futher research should focus on clarifying the mechanistic diversity of Gal-9 signaling in different cellular contexts and evaluating the therapeutic potential of modulating Gal-9–dependent neutrophil-T cell interactions. Such strategies may yield new insights into the control of inflammation, immune tolerance, and tumor progression.
Author Response
- This integrative review presents a compelling and timely synthesis of neutrophil–T-cell interactions across diverse pathological contexts, including adult T-cell leukemia/lymphoma (ATL), Sézary syndrome (SS), COVID-19, and psoriasis. The authors successfully highlight the evolving understanding of neutrophils—from traditionally short-lived antimicrobial effectors to versatile regulators of adaptive immunity and contributors to disease progression. By positioning neutrophils and the neutrophil-derived galectin-9 (Gal-9) axis as conserved drivers of immune modulation, the review contributes meaningfully to current immunological discourse.
Reply
We sincerely thank Reviewer 2 for their thoughtful, encouraging, and comprehensive evaluation of our manuscript. We appreciate the reviewer’s recognition of the conceptual novelty of the neutrophil–Galectin-9 (Gal-9) axis and the interdisciplinary synthesis across oncology, dermatology, and infectious disease.
- A major strength of the study is its innovative methodology, which combines AI-assisted literature synthesis using GPT-5 and Adobe Acrobat AI Assistant with expert human validation. This hybrid approach not only accelerates data integration across large temporal (2000–2025) and thematic scales but also demonstrates a promising model for future scholarly work, where artificial intelligence augments but does not replace expert interpretation. The clarity with which the authors delineate this workflow enhances the transparency and reproducibility of the review.
We are also grateful for the positive assessment of our AI-assisted methodology and the hybrid workflow combining GPT-5, Adobe Acrobat AI Assistant, and expert human validation.
A sentence (lines 71-73) was added and
Table 1 was created to clarify the role of AI, and its roles were also described (lines 86-88).
- Identifying persistent neutrophil activation and elevated Gal-9 expression as shared immunological signatures provides an important conceptual bridge linking innate dysregulation to downstream T-cell dysfunction. The discussion of how Gal-9 may mediate TCR signaling alterations in ATL/SS, as well as its association with T-cell exhaustion and IL-17–driven inflammation in COVID-19 and psoriasis, is both biologically plausible and clinically relevant. This unified framework has clear implications for therapeutic development, potentially informing strategies targeting neutrophil activation, Gal-9 signaling, or the neutrophil–T-cell interface more broadly. Overall, the paper reflects a well-organized, novel, and forward-looking review. As the authors state at the end, futher research should focus on clarifying the mechanistic diversity of Gal-9 signaling in different cellular contexts and evaluating the therapeutic potential of modulating Gal-9–dependent neutrophil-T cell interactions. Such strategies may yield new insights into the control of inflammation, immune tolerance, and tumor progression.
Reply
We are grateful for this interpretation. We have added a short integrative statement in the Conclusion to make this conceptual bridge more explicit while remaining cautious about overgeneralization (lines 284-289).

Reviewer 4 Report
Comments and Suggestions for Authors
The idea behind this work, which explores the feasibility of combining AI-driven literature synthesis with expert review to uncover unifying immunological mechanisms and therapeutic targets across malignancy and inflammation, is unclear. Does it actually refer to the use of these AI tools to identify or propose new ways of understanding the regulation of a biological effect? The objective is not very clear, and this is not achieved in the content and conclusions of the work. Would this tool offer any advantages over a meta-analysis? Could it be considered as a methodological alternative?
It is interesting to identify the fundamental role of Galectin-9, particularly in different study models. This work raises aspects related to the role of Galectin-9 in modulating neutrophil function, comparing it to various broad pathological processes, from viruses to tumors. The involvement of neutrophils in the activation of antigen processing and presentation via MHC is not considered, and basing this on the simple capacity of a potential cofactor like Galectin-9 should be discussed, as this will determine the permissive or non-permissive capacity of this cell group, which is the most abundant and acts immediately in processes of acute inflammatory activation.
Some models of autoimmunity do not allow us to identify the authors' objective, for example, in psoriasis. IL-17 is fundamental in psoriasis, as it initiates and perpetuates the inflammatory response by acting on keratinocytes to cause cell proliferation and the production of pro-inflammatory molecules. This cytokine is produced by cells such as Th17, γδ cells, and ILC3. How does the Galectin-9 and neutrophil relationship influence this?
Minor comment
The statement "Neutrophils, once viewed as short-lived antimicrobial cells, are now recognized as key regulators of adaptive immunity and tumor progression" is an anachronistic idea.
Author Response
Reviewer 4
- The idea behind this work, which explores the feasibility of combining AI-driven literature synthesis with expert review to uncover unifying immunological mechanisms and therapeutic targets across malignancy and inflammation, is unclear. Does it actually refer to the use of these AI tools to identify or propose new ways of understanding the regulation of a biological effect? The objective is not very clear, and this is not achieved in the content and conclusions of the work. Would this tool offer any advantages over a meta-analysis? Could it be considered as a methodological alternative?
Reply
We thank the reviewer for carefully reading our manuscript and for providing insightful comments, particularly regarding the conceptual positioning of our AI-assisted approach and the immunobiological role of Galectin-9 and neutrophils in different disease models.
We have revised the manuscript to clarify the objectives, better situate our methodology relative to traditional approaches such as meta-analysis, and refine the discussion of antigen presentation and psoriasis/IL-17 biology.
Below we respond to each comment point-by-point.
The AI assisted system accelerates the mapping and integration of a very broad and heterogeneous literature (multiple diseases, >20 years). Ultimately, I chose the option that aligned with the main purpose of this paper. (lines 90-92, Table 2). Furthermore two figures were created by Chat-GPT (Figures 1 and 2).
- It is interesting to identify the fundamental role of Galectin-9, particularly in different study models. This work raises aspects related to the role of Galectin-9 in modulating neutrophil function, comparing it to various broad pathological processes, from viruses to tumors. TSome models of autoimmunity do not allow us to identify the authors' objective, for example, in psoriasis. IL-17 is fundamental in psoriasis, as it initiates and perpetuates the inflammatory response by acting on keratinocytes to cause cell proliferation and the production of pro-inflammatory molecules. This cytokine is produced by cells such as Th17, γδ cells, and ILC3. How does the Galectin-9 and neutrophil relationship influence this?
Reply
- Use this hybrid process to reveal a unifying conceptual axis—the neutrophil–Gal-9–T-cell interface—across ATL, SS, COVID-19, and psoriasis, which are rarely discussed together. In the Introduction , we have explicitly defined our objective as:“to illustrate how AI-assisted literature synthesis can support expert-guided identification of convergent immunological themes—here, a neutrophil–Gal-9–T-cell axis—across malignant and inflammatory diseases, rather than to perform a formal meta-analysis.”(line 64-6, figures 1 and 2)
- In the Methods section, we have clarified that:
A: Our approach is most closely aligned with a systematized scoping review enhanced by AI tools for retrieval, clustering, and summarization.(lines 80-886, Table,)
B:AI is used to screen, organize, and summarize candidate literature, while all inclusion/exclusion decisions, interpretation of mechanisms, and final wording remain under human expert control.(lines 87-92, Table1)
- I have described the importance of IL-17 in PS and its interplay with Gal-9 was described in PS.(lines 206-218)
- The involvement of neutrophils in the activation of antigen processing and presentation via MHC is not considered, and basing this on the simple capacity of a potential cofactor like Galectin-9 should be discussed, as this will determine the permissive or non-permissive capacity of this cell group, which is the most abundant and acts immediately in processes of acute inflammatory activation.
Reply We thank the reviewer for this insightful comment. While neutrophils can acquire antigen-processing and MHC-dependent antigen-presenting functions under specific conditions, this review did not aim to address this aspect comprehensively. Instead, we focused on the role of neutrophils as rapid and abundant innate responders that modulate T-cell signaling environments, particularly via Galectin-9. The antigen-presenting capacity of neutrophils is highly context-dependent and not uniformly permissive across disease states. Accordingly, Galectin-9 is discussed here as a modulatory factor influencing T-cell activation thresholds rather than as a determinant of classical MHC-dependent antigen presentation. We have clarified this limitation in the revised Discussion.
4
The statement "Neutrophils, once viewed as short-lived antimicrobial cells, are now recognized as key regulators of adaptive immunity and tumor progression" is an anachronistic idea.
Reply
We appreciate this observation. Our intention was to provide a brief historical perspective; however, we agree that the wording could suggest an oversimplified or outdated dichotomy.
We have therefore revised the sentence (lines 31-2)
This phrasing better acknowledges that the expanded view of neutrophils has been developing over many years and is now well established in contemporary immunology.

Round 2
Reviewer 1 Report
Comments and Suggestions for Authors
After careful review, I confirm that the manuscript has been sufficiently improved to warrant publication in Reports.
Author Response
Reviewer 2
Some of the comments were addressed with limitations. Following comments still need consideration.
- Line#71: ‘A major strength of the study is its innovative methodology… this hybrid approach not only accelerates data integration but also demonstrates a promising model where artificial intelligence augments but does not replace expert interpretation’ it looks fragmented (methodology followed by three dots…authors wants to add something there? Reframe appropriately)
Reply;
We agree with the reviewer’s observation. The sentence has been rewritten to remove the ellipsis and improve clarity and grammatical completeness (lines 71-74)
- There are typos issues in the manuscript e.g. spacing, comma, full stop, double dots (“)etc. Proofreading required.
Reply: We have carefully proofread the entire manuscript and corrected typographical and formatting issues, including spacing, punctuation, duplicated symbols, and inconsistent use of commas and periods. These corrections were made throughout the text without changing the scientific content.
- Figure 1 legends are poorly defined. Merely reference cited.
Reply
We appreciate this comment and have revised the Figure 1 legend to provide clearer explanatory descriptions for each disease context. The legend now explicitly summarizes the biological roles of neutrophils, Galectin-9, and T-cell receptor modulation in each condition, rather than only citing references. This revision improves figure readability and standalone interpretability.
- Figure 2. TCR downregulation is followed by full stop ?
I have corrected the point you mentioned. (Figure 2)

Reviewer 2 Report
Comments and Suggestions for Authors
Comments for Authors:
Some of the comments were addressed with limitations. Following comments still need consideration.
1. Line#71: ‘A major strength of the study is its innovative methodology… this hybrid approach not only accelerates data integration but also demonstrates a promising model where artificial intelligence augments but does not replace expert interpretation’ it looks fragmented (methodology followed by three dots…authors wants to add something there? Reframe appropriately)
2. There are typos issues in the manuscript e.g. spacing, comma, full stop, double dots (“)etc. Proofreading required.
3. Figure 1 legends are poorly defined. Merely reference cited.
4. Figure 2. TCR downregulation is followed by full stop ?
Author Response

(The authors gave the same response as above.)

Reviewer 4 Report
Comments and Suggestions for Authors
The revised manuscript is much improved. In general, the authors have addressed my concerns.

Author Response
The revised manuscript is much improved. In general, the authors have addressed my concerns.
Reply;
Thanks to your comments, my paper has improved.